# Retinal structure and related factors in 8-year-old Japanese children: The Yamanashi adjunct study of the Japan Environment and Children's Study

Ryo Harada[1,2], Mingxue Bao[1], Natsuki Okabe[1], Yuka Kasai[1], Airi Takahashi[1], Chio Kuleshov[1], Yumi Shigemoto[1], Tadao Ooka[3], Hiroshi Yokomichi [4], Kunio Miyake[4], Reiji Kojima[4], Ryoji Shinohara[5], Hideki Yui[5], Sanae Otawa[5], Anna Kobayashi[5], Megumi Kushima[5], Zentaro Yamagata[5], Kenji Kashiwagi [1]*, The Yamanashi Adjunct Study of the Japan Environment and Children's Study Group[¶]

1 Department of Ophthalmology, Interdisciplinary Graduate School of Medicine, University of Yamanashi, Yamanashi, Japan, 2 Department of Ophthalmology, Kofu Kyoritsu Clinic, Yamanashi, Japan, 3 Department of Health Sciences, Interdisciplinary Graduate School of Medicine, University of Yamanashi, Yamanashi, Japan, 4 Department of Epidemiology and Environmental Medicine, Interdisciplinary Graduate School of Medicine and Engineering, University of Yamanashi, Yamanashi, Japan, 5 Center for Birth Cohort Studies, Interdisciplinary Graduate School of Medicine, University of Yamanashi, Yamanashi, Japan

¶Membership of the Yamanashi Adjunct Study of the Japan Environment and Children's Study Group is provided in the Acknowledgments.
* kenjik@yamanashi.ac.jp

## Abstract

### Purpose

The purpose of this study was to investigate the relationships among retinal/choroidal structures and associated factors in 8-year-old Japanese children who participated in the Yamanashi Adjunct Study of the Japan Environment and Children's Study (JECS).

### Methods

This was a study of 557 8-year-old Japanese children (274 boys, 283 girls) who participated in the JECS at the University of Yamanashi from June 2021 to March 2022. The visual acuity, spherical equivalent (SE), axial length (AL), and body height of the participants were measured. Retinal and choroidal thickness were measured using spectral domain optical coherence tomography (NIDEK RS-3000 Advance, Gamagori, Japan).

### Results

This study included 304 participants (150 boysand 154 girls). The mean values were 23.08 ± 0.75mm for AL, −0.35 ± 0.79D for SE, and 0.08 ± 0.20 for uncorrected logMAR.The mean retinal thickness was 309.7 ± 10.9μm for all retinal layers (ARLs), 220.9 ± 15.3μm for the fovea, 134.8 ± 6.8μm for the outer retina, 76.1 ± 4.5μm for the

**Data availability statement:** Data are unsuitable for public deposition due to ethical restrictions and legal framework of Japan. It is prohibited by the Act on the Protection of Personal Information (Act No. 57 of 30 May 2003, amendment on 9 September 2015) to publicly deposit the data containing personal information. Ethical Guidelines for Medical and Health Research Involving Human Subjects enforced by the Japan Ministry of Education, Culture, Sports, Science and Technology and the Ministry of Health, Labour and Welfare also restricts the open sharing of the epidemiologic data. All inquiries about access to data should be sent via email (cbcs-yamanashi-as@yamanashi.ac.jp). Inquiries sent to this email address will be handled by the Koshin Unit Center of JECS.

**Funding:** This adjunct study was supported by Japan Society for the Promotion of Science, No.23K09003. The funder had no role in study design, data collection, and analysis, decision to publish, and preparation of the manuscript.

**Competing interests:** None.

inner retina, and 29.1±2.1μm for the nerve fiber layer. The mean choroidal thickness was 301.5±34.0μm. Multivariable analysis revealed that the thickness of ARLs had a significant negative correlation with AL (β=−0.20, p<0.001) and a significant positive correlation with body height (β=0.11, p=0.04), with males having a greater retinal thickness than females (β=−0.22, p<0.001). Foveal thickness was significantly positively correlated with body height (β=0.13, p=0.02) but not with AL (β=0.11, p=0.06) or sex (β=−0.04, p=0.55).

## Conclusion

The structures of the retina and choroid in 8-year-old Japanese children were associated with several factors, but the associations varied by retinal site and layer.

---

## Introduction

Optical coherence tomography (OCT) is a useful technique for evaluating retinal structures in children because it can noninvasively acquire *in vivo* images of the retina in a short time [1]. OCT provides clinically valuable data for the diagnosis and monitoring of pediatric retinal diseases such as retinopathy of prematurity and retinal dystrophy, as well as optic nerve diseases [2–4]. It has been reported that the foveal structure is complete by approximately 18 months of age based on *in vivo* OCT images [5]. Recent studies on the analysis of retinal structure in children using OCT have progressed, and the development of the fovea and each layer of the posterior retina from infancy to school age has been elucidated. Furthermore, studies have shown that changes in retinal structure between the fovea and the peripheral retina vary with growth and that the increase in retinal thickness in the fovea due to photoreceptor cell elongation continues until approximately puberty [6,7].

Myopia is more likely to occur at school age [8,9], and it has been documented that retinal thickness decreases as axial length (AL) increases [10]. A longitudinal study in children reported that while the retinal thickness changed slightly with age, the effects of refractive error and AL were more significant [11]. Although some reports have shown that as AL increases, the risk of myopia increases and the thickness of some retinal layers significantly decreases [12–14], other studies have not shown a similar effect [15]. In children, however, mechanical extension may not be the only factor involved in the change in retinal thickness since growth and development may also play a role. The databases installed in OCT devices rarely contain normative data on children under 18 years of age, and there have been no large-scale studies on Japanese children. Therefore, it is important to investigate what factors affect retinal structure in children to evaluate the development of pediatric retinal structure and to establish normal reference values for children.

The purpose of this study was to examine the relationships between retinal thickness and AL, refractive error, and body height in 8-year-old Japanese children.

## Materials and methods

### Study design and participants

This study was part of the Yamanashi Adjunct Study of the Japan Environment and Children's Study (JECS) [9,16,17]. The JECS is a national project of the Ministry of the Environment, an ongoing nationwide birth cohort study that aims to determine the impact of environmental factors on child health from the fetal period until the child is grown. More than 100,000 pregnant women enrolled in the 15 JECS study regions across Japan between January 2011 and March 2014 are included in the study. The JECS protocol and baseline data are as previously reported [18,19]. For the present study, participants were 559 8-year-old Japanese children (277 boys and 282 girls) who participated in the Yamanashi Adjunct Study of JECS at the University of Yamanashi from June 2021 to March 2022. The JECS protocol was reviewed and approved by the Ministry of the Environment's Institutional Review Board on Epidemiological Studies. This study was approved by the Ethical Review Committee of the University of Yamanashi School of Medicine and was conducted in accordance with the Declaration of Helsinki. Written consent was obtained from all the participants and their parents.

### Examinations

In the Yamanashi Adjunct Study of JECS, physical measurements, physical fitness tests, and blood tests were conducted on the same day as the ophthalmologic examination. Ophthalmologic examinations included measuring uncorrected visual acuity, corrected visual acuity, AL, and objective refraction, as described in the study by Okabe *et al* [9]. In addition, OCT measurements of macular retinal thickness were also performed in this study. Macular retinal thickness was measured using spectral domain optical coherence tomography (NIDEK RS-3000Advance2 SD-OCT, Gamagori, Japan). All ophthalmologic examinations were performed by ophthalmologists and orthoptists.

### OCT measurements

The RS-3000 Advance2 SD-OCT scans at 85,000 A-scans per second, with a light source wavelength of 880 nm and a resolution of 20 μm in the lateral direction and 7 μm in the depth direction. A macular radial scan with 6 lines of 6-mm wide scan lines every 30° was used to measure macular retinal thickness (Fig 1a). The macular retina was analyzed and grouped into five layers (Fig 1b): Nerve fiber layer (NFL) thickness from the internal limiting membrane (ILM) to the NFL; Ganglion cell-inner plexiform layer (GC-IPL) thickness from the ganglion cell layer (GCL) to the inner plexiform layer (IPL); Outer nuclear layer (ONL) to Bruch's membrane (BM) thickness; Fovea. The average macular thickness was then defined as IML to BM. The choroid was measured using the border between the sclera and the choroid as the boundary between the lower part of the OCT image, where there is no image depiction, and the area where there is an image depiction, albeit indistinct.

The semiautomatic segmentation program of NAVIS-EX image analysis software was used to analyze each layer. In cases of segmentation errors, manual correction was performed by one of the examiners (HR). The Early Treatment Diabetic Retinopathy Study (ETDRS) grid was used for the analysis of the macular retina. The central foveal ring with a 1-mm diameter was defined as the central sector, the adjacent inner sector was defined as the section from 1 to 3 mm from the center, and the outer sector was defined as the section from 3 to 6 mm from the center (Fig 1c). Average macular thickness is measured BM from the ILM and then averaged over the entire 6 mm circle. To correct for the ocular magnification effect [20], we used NAVIS-EX onboard AL correction and analyzed the images after inputting the AL values of each examinee.

### Inclusion criteria

The inclusion criteria for participants were as follows: (1) a corrected visual acuity of logMAR 0 or better; (2) no strabismus, amblyopia, or aniseikonia. Further inclusion criteria for eye imaging characteristics included the following: (1) a

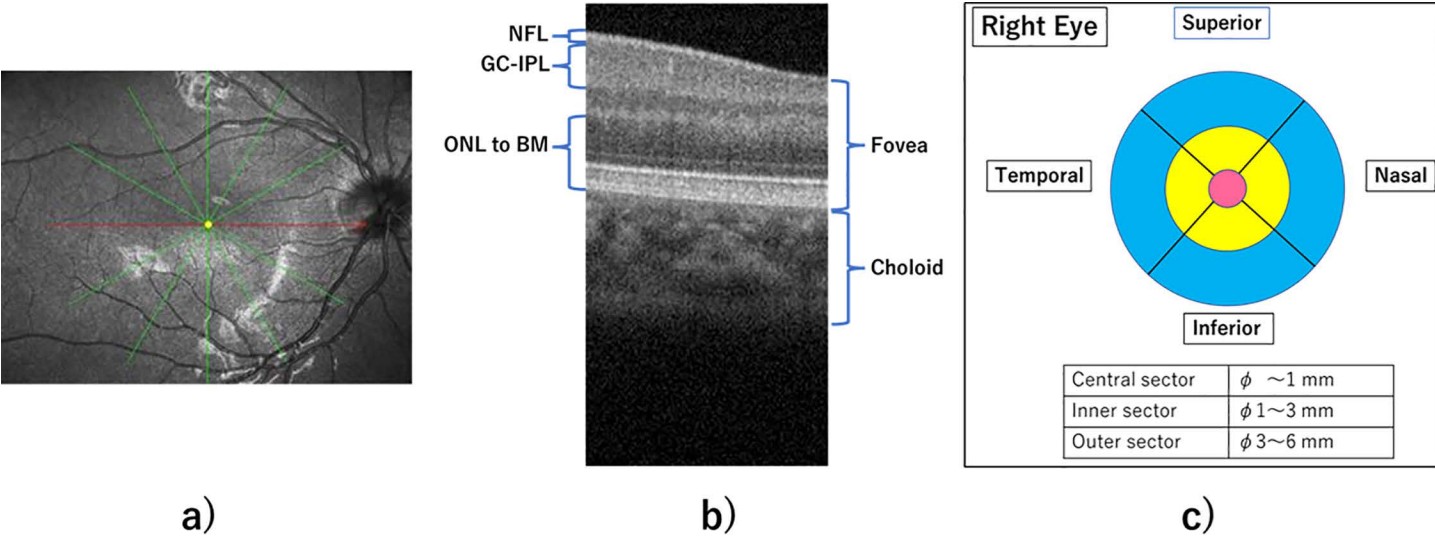

**Fig 1. Definition of OCT measurement. a)** Macular radial scan, **b)** Retinal layers used in the OCT measurement: Nerve fiber layer (NFL) thickness from the internal limiting membrane (ILM) to the NFL; Ganglion cell-inner plexiform layer (GC-IPL) thickness from the ganglion cell layer (GCL) to the inner plexiform layer (IPL); Outer nuclear layer (ONL) to Bruch's membrane (BM) thickness; Fovea; Choroid. The average macular thickness was then defined as IML to BM **c)** Diagram of the nine ETDRS subfields.

signal-to-noise ratio of the AL measurement of 5.0 or greater, (2) a scan signal index (SSI) of the OCT image greater than 7, (3) a scan quality index (SQI) greater than 2, (4) central fixation, and (5) all 6 lines obtained in the radial scan image. By convention, the right eye was used for measurement, and only when the right eye was unavailable was the left eye used. Twenty-one children with a visual acuity less than logMAR 0 and 17 children with strabismus, amblyopia, or aniseikonia were initially excluded. Additionally, 5 children who did not cooperate with OCT measurements and AL imaging and 212 children who had poor image quality due to blinking, poor fixation, or poor postural maintenance were excluded. In total, 304 eyes of 304 participants (150 boys and 154 girls) met the inclusion criteria. Of the 304 who met the recruitment criteria, 186 used their right eye and 118 used their left eye. On OCT images, the SSI and SQI of the 304 participants included in the analysis were 8.8±1.1 and 4.1±1.1, respectively. A comparison of the data from the included and excluded groups revealed that the excluded group had a significantly longer AL, more cases of myopia, and significantly poorer uncorrected visual acuity (S1 Table in S1 File). The choroid was included in the analysis of 239 (117 males and 122 females) of 304 participants who underwent measurements.

## Statistical analysis

EZR for Windows (Jichi Medical University, Japan) [21] was used for the statistical analysis of the relationships between macular retinal and choroidal thickness and AL, spherical equivalent (SE), and body height. Visual acuity, AL, SE, and body height were analyzed using t-test for sex differences. Comparisons between quadrants were made using Bonferroni correction with one-way analysis of variance of the four sectors—temporal, superior, nasal, inferior—measured by OCT. Single regression analysis between AL, SE and macular retinal and choroidal thickness was performed with Pearson's product-moment correlation coefficient. Multivariable regression analysis was performed with macular retinal and choroidal thickness as dependent variables and AL, sex, and body height as independent variables. Differences in visual acuity, AL, SE, and body height between the included and excluded groups were analyzed using t tests. The boy/girl ratio of the included and excluded groups was analyzed using the chi-square test. The difference between the OCT measurements

of the left and right eyes of the included groups was analyzed using the Mann–Whitney U test. Two-sided p-value of less than 5% were considered to indicate statistical significance. The measurement results are shown as the mean ± standard deviation.

## Results

### Demographics

The mean values for AL, SE, uncorrected logMAR, and body height of the 304 participants are shown in Table 1. Boys had significantly longer AL (P<0.001), and there were no significant sex differences in SE, logMAR, or body height.

### Mean Thickness of the Retinal Layers and Choroid

The mean thicknesses of the macular retinal layers and the choroid are summarized in Table 2. The thickest retinal layer measurement was found for the fovea, followed by the ONL to BM and the GC-IPL. The thinnest measurements were found for the NFL.

### Results for each retinal layer

The mean thicknesses of the Average macular thickness, ONL to BM, GC-IPL, and NFL per quadrant are summarized in Table 3. Each quadrant included the measurements for the inner and outer sectors. The Average macular thickness showed significant differences in all quadrants (P<0.001). The ONL to BM thickness significantly differed among the four quadrants (P<0.05). For the GC-IPL, only the nasal quadrant was significantly thicker than the other sectors (p<0.001).

**Table 1. Average axial length, spherical equivalent, refraction, and body height of the included participants.**

|  | All participants (n = 304) | Boys(n = 150) | Girls(n = 154) | Pª |
|---|---|---|---|---|
| AL (mm) | 23.06 ± 0.74 (21.95-24.31) | 23.30 ± 0.72 (22.25-24.47) | 22.82 ± 0.68 (21.74-24.03) | <0.001 |
| SD (D) | −0.32 ± 0.77 (−1.76- + 0.63) | −0.35 ± 0.77 (−1.63- + 0.58) | −0.30 ± 0.77 (−1.60- + 1.25) | 0.52 |
| Uncorrected logMAR | 0.07 ± 0.19 (0.00-0.52) | 0.07 ± 0.19 (0.00-0.52) | 0.07 ± 0.18 (0.00-0.52) | 0.90 |
| Body height (cm) | 125.2 ± 5.1 (117.9-133.4) | 125.1 ± 5.2 (116.9-133.2) | 125.3 ± 4.9 (118.2-133.9) | 0.75 |

The data are presented as the means ± standard deviations (5th–95th percentiles).

Axial Length (AL), Spherical Equivalent (SD).

ª P values are sex differences by Independent samples t-test.

**Table 2. Summary of the mean retinal thickness values per layer and the mean choroidal thickness.**

| OCT Measurements | Thickness(μm) |
|---|---|
| Average macular thickness | 309.7 ± 10.9(280–340) |
| Fovea | 220.9 ± 15.3(180–274) |
| ONL to BM | 134.8 ± 6.8(114–153) |
| GC-IPL | 76.1 ± 4.5(63–90) |
| NFL | 29.1 ± 2.1(24–36) |
| Choroid | 301.5 ± 34.0(196–377) |

The data are presented as the means ± standard deviations (5th–95th percentiles).

n: 304 (Excluding the choroid. The choroid was measurable 239 eyes).

The NFL also significantly differed among all quadrants (P < 0.01). The results of the one-way ANOVA are shown in S2 Table in S1 File. The values for each of the nine sectors are shown in S3, S4, S5, S6, and S7 Tables in S1 File.

## Analysis of factors associated with retinal layer structure by layer

Foveal thickness was significantly positively correlated with AL (r = 0.14, P = 0.01) but not with the SE (r = −0.04, p = 0.50). Choroidal thickness was not significantly correlated with AL (r = −0.11, p = 0.10) and was significantly positively correlated with the SE (r = 0.14, p = 0.02).

Single regression analysis of AL and the SE for the whole 6-mm circle and the central, inner, and outer sectors (all averaged among the 4 quadrants) are summarized in Tables 4 and 5. AL was significantly negatively correlated with the whole (r = −0.11, p = 0.04), inner (r = −0.13, p = 0.02), and outer sectors (r = −0.12, p = 0.03) for average macular thickness

**Table 3. Summary of the mean retinal thickness values per layer and per quadrant.**

| OCT Measurements | Temporal (n = 304) | Superior (n = 304) | Nasal (n = 304) | Inferior (n = 304) |
|---|---|---|---|---|
| Thickness(μm) Average macular thickness | 304.9 ± 11.3 (288-323) | 319.5 ± 11.9 (301-340) | 328.6 ± 12.4 (310-351) | 311.8 ± 11.5 (294-332) |
| ONL to BM | 135.4 ± 7.5 (122-147) | 129.9 ± 8.3 (117-144) | 132.4 ± 9.1 (118-148) | 128.1 ± 7.9 (115-140) |
| GC-IPL | 75.1 ± 4.9 (67-84) | 75.4 ± 4.8 (68-83) | 79.4 ± 4.8 (72-88) | 74.6 ± 4.6 (68-83) |
| NFL | 18.7 ± 2.2 (15-22) | 32.2 ± 3.0 (28-38) | 34.2 ± 2.9 (30-39) | 31.4 ± 2.8 (28-37) |

The data are presented as the means ± standard deviations (5th–95th percentiles).

**Table 4. Correlation between retinal layer thickness and AL.**

| | Sector | | | | | | | |
|---|---|---|---|---|---|---|---|---|
| | Whole | | Central | | Inner | | Outer | |
| | r | P | r | P | r | P | r | P |
| Average macular thickness | −0.11 | 0.04 | 0.04 | 0.51 | −0.13 | 0.02 | −0.12 | 0.03 |
| ONL to BM | −0.08 | 0.19 | 0.02 | 0.74 | −0.10 | 0.07 | −0.07 | 0.23 |
| GC-IPL | −0.10 | 0.08 | – | – | −0.13 | 0.02 | −0.04 | 0.44 |
| NFL | −0.08 | 0.19 | – | – | 0.03 | 0.59 | −0.11 | 0.05 |

Correlation coefficient (r) and P value (P) of Pearson's product-moment correlation coefficient.

**Table 5. Correlation between retinal layer thickness and the SE.**

| | Sector | | | | | | | |
|---|---|---|---|---|---|---|---|---|
| | Whole | | Central | | Inner | | Outer | |
| | r | P | r | P | r | P | r | P |
| Average macular thickness | 0.08 | 0.18 | −0.02 | 0.70 | 0.07 | 0.22 | 0.10 | 0.09 |
| ONL to BM | −0.08 | 0.14 | −0.09 | 0.10 | −0.07 | 0.20 | 0.08 | 0.17 |
| GC-IPL | 0.12 | 0.03 | – | – | 0.12 | 0.03 | 0.11 | 0.06 |
| NFL | 0.01 | 0.87 | – | – | 0.08 | 0.17 | 0.05 | 0.35 |

Correlation coefficient (r) and P value (P) of Pearson's product-moment correlation coefficient.

and the inner sector (r=−0.13, p=0.02) for the GC-IPL. The SE was significantly positively correlated with the whole sector (r=0.12, p=0.03) and inner sector (r=0.12, p=0.03) for the GC-IPL.

## Analysis of factors associated with retinal structure by site

A single regression analysis of the association of nine subfields in each layer with AL and the SE is illustrated in Fig 2. The outer superior (r=−0.13, p=0.02), outer nasal (r=−0.19, p=0.001), inner superior (r=−0.17, p=0.003), and inner nasal subfields (r=−0.16, p=0.006) of average macular thickness had significant negative correlations with AL, while the outer nasal subfield (r=0.14, p=0.01) had a significant positive correlation with the SE. The outer superior (r=−0.11, p=0.04) and inner superior (r=−0.15, p=0.007) subfields of the ONL to BM showed a significant negative correlation with AL, while the outer temporal (r=−0.13, p=0.02), inner temporal (r=−0.15, p=0.007) and inner inferior (r=−0.11, p=0.04) subfields showed a significant negative correlation with the SE. The inner superior (r=−0.12, p=0.04), inner nasal (r=−0.17, p=0.003), and inner inferior (r=−0.13, p=0.02) subfields of the GC-IPL showed significant negative correlations with AL, while the outer nasal (r=0.13, p=0.02), outer inferior (r=0.19, p=0.001), and inner nasal (r=0.14, p=0.01) subfields showed a significant positive correlation with the SE. The outer nasal (r=−0.19, p=0.001) and inner superior (r=−0.12, p=0.03) subfields of the NFL showed a significant negative correlation with AL, while the inner inferior subfield (r=−0.14, p=0.01) had a significant negative correlation with the SE.

Table 6 shows the results of multivariable regression analysis with the thickness of each layer as the dependent variable and AL, sex, and body height as the independent variables. Sex was analyzed by assigning boys a value of 0 and girls a value of 1. Scatter plots of the average macular thickness in all sectors versus AL and body height are shown in S8 Fig in S1 File. Foveal thickness showed a significant positive correlation with body height (β=0.13, p=0.02) but not with AL or sex. The whole sector of average macular thickness showed a significant negative correlation with AL (β=−0.20, p<0.001) and sex (β=−0.22, p<0.001) and a significant positive correlation with body height (β=0.11, p=0.04). The central sector of average macular thickness showed a significant negative correlation with sex (β=−0.14, p=0.01) and a significant positive correlation with body height (β=0.14, p=0.01) but no correlation with AL. The inner sector of average macular thickness showed a significant negative correlation with AL (β=−0.22, p<0.001) and sex (β=−0.25, p<0.001) but no correlation with body height. The outer sector of the average macular thickness had a significant negative correlation with AL (β=−0.18, p=0.002) and sex (β=−0.14, p=0.01) but no correlation with body height. The choroid was not significantly correlated with AL, sex, or body height. The analyses of the ONL to BM, GC-IPL, and NFL are shown in S9, S10, and S11 Tables in S1 File.

## Discussion

This is the cross-sectional study of Japanese children at a fixed age of 8 years in which the thicknesses of the average macular thickness, ONL to BM, GC-IPL, and NFL, fovea, and choroid were measured. Unlike previous reports, this study included not only ocular parameter but also body height measurements. The mean thickness of the fovea, as well as at the whole and central sectors of average macular thickness, increased with increasing body height. On the other hand, the whole, inner, and outer sectors of average macular thickness decreased with increasing AL. Retinal thickness was greater in boys. Choroidal thickness was positively correlated with refractive error; however, in this study, retinal thickness had a weak positive association with refractive error.

Table 7 compares the results of this study with those of previous reports. Although a reliable comparison could not be made because of the differences in the OCT machines used for the measurements, the results for the retinal thickness of children were similar. Comparisons of the values reported in the present study with those of Passani *et al*., who assessed subjects up to 18 years of age, revealed similar values; however, differences in retinal thickness as influenced by race should also be considered[14]. This racial influence was previously reported in the studies of Huynh *et al*. [22], and El Dairi *et al*. [23] Bario-Bario *et al.* reported a significant increase in average macular thickness with increasing age

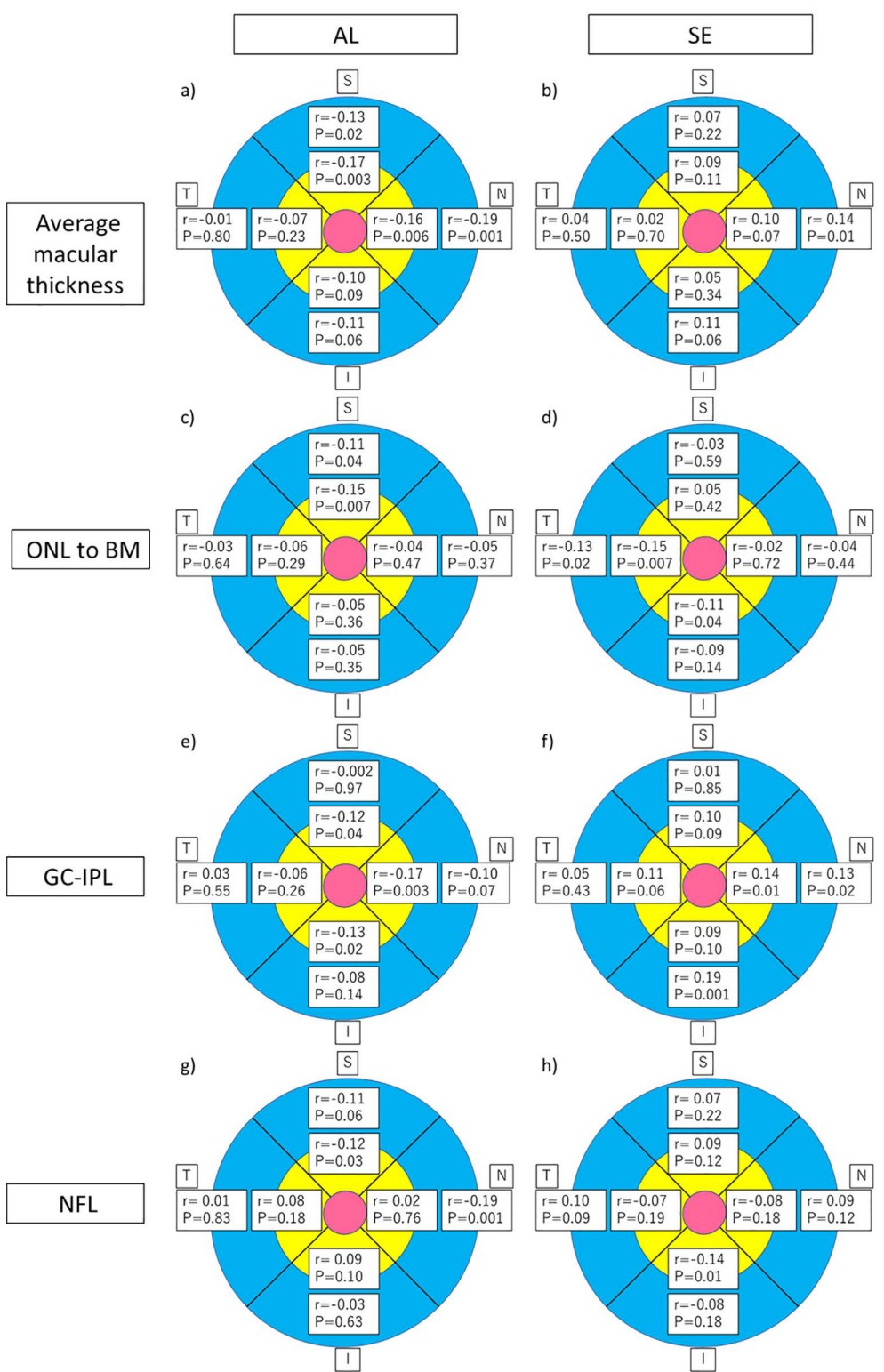

**Fig 2. Correlations between retinal layer thickness and AL and SE.** Correlation coefficient (r) and p value (P) of Pearson's product-moment correlation coefficient. The quadrants are the temporal (T), superior (S), nasal (N), and inferior (I) subfields. a) average macular thickness vs. AL, b) average macular thickness vs. SE, c) ONL to BM vs. AL, d) ONL to BM vs. SE, e) GC-IPL vs. AL, f) GC-IPL vs. SE, g) NFL vs. AL, h) NFL vs. AL.

Table 6. Multivariable regression analysis of the relationships between retinal and choroidal thickness and axial length, sex, and body height.

| Thickness | AL | | | | | Sex | | | | | Body height | | | | |
|---|---|---|---|---|---|---|---|---|---|---|---|---|---|---|---|
| | B [a] | 95%CI of B | | β [b] | P | B [a] | 95%CI of B | | β [b] | P | B [a] | 95%CI of B | | β [b] | P |
| Whole [c] | −2.99 | −4.73 | −1.25 | −0.20 | <0.001 | −4.73 | −7.27, | −2.19 | −0.22 | <0.001 | 0.25 | 0.01 | 0.49 | 0.11 | 0.04 |
| Central [c] | −0.66 | −3.22 | 1.90 | −0.03 | 0.61 | −4.54 | −8.27 | −0.81 | −0.14 | 0.01 | 0.43 | 0.08 | 0.78 | 0.14 | 0.01 |
| Inner [c] | −3.70 | −5.64 | −1.76 | −0.22 | <0.001 | −6.18 | −9.01 | −3.35 | −0.25 | <0.001 | 0.20 | −0.06 | 0.47 | 0.08 | 0.13 |
| outer [c] | −2.85 | −4.72 | −0.98 | −0.18 | 0.002 | −3.32 | −6.04 | −0.59 | −0.14 | 0.01 | 0.24 | −0.02 | 0.50 | 0.11 | 0.06 |
| Fovea | 2.32 | −0.15 | 4.80 | 0.11 | 0.06 | −1.08 | 4.69, | 2.54 | −0.04 | 0.55 | 0.38 | 0.04 | 0.73 | 0.13 | 0.02 |
| Choroid | −5.57 | −11.98 | 0.83 | −0.12 | 0.08 | −1.63 | −11.04 | 7.79 | −0.02 | 0.73 | 0.32 | −0.55 | 1.19 | 0.05 | 0.46 |

Sex was analyzed by assigning boys a value of 0 and girls a value of 1.

[a] Nonstandardized Regression Coefficient B.

[b] Standardized Regression Coefficient β.

[c] Average macular thickness.

Table 7. Comparison of the average macular thickness per sector in the present study with previous reports.

| | Present Study | Al-Haddad et al. (2014) [12] | Bario-Bario et al. (2013) [13] | Passani et al. (2019) [14] | Normative database |
|---|---|---|---|---|---|
| Age | 8 | 6-18 | 4-17 | 5-18 | 20-29 |
| Participants | 304 | 108 | 281 | 94 | |
| Equipment | RS-3000 Advance 2 | Cirrus-HD | Cirrus 4000 | Spectralis | RS-3000 Advance 2 |
| Thickness(μm) Whole sector | 311.1 (281-343) | NA | 283.6 (260.1-308.0) | NA | NA |
| Central sector | 253.9 (208-295) | 280 (259–298) | 253.8 (220.1-287.4) | 274.96 (247.80-308.20) | 261.2(235-287) |
| Inner Temporal | 325.7 (290-370) | 309 (282–337) | 311.0 (284.6-338.0) | 331.55 (313.60-352.40) | 334.5 (314-354) |
| Inner Superior | 341.2 (307-378) | 321 (294–348) | 317.6 (279.1-351.8) | 344.15 (324.40-365.40) | 349.7 (328-371) |
| Inner Nasal | 339.8 (302-381) | 321 (294–347) | 324.8 (298.1-354.0) | 346.22 (325.80-371.20) | 348.2 (325-371) |
| Inner Inferior | 330.0 (295-380) | 318 (293–343) | 319.3 (290.1-347.0) | 341.51 (320.80-368.40) | 344.6 (323-365) |
| Outer Temporal | 289.7 (253-331) | 263 (242–287) | 268.0 (243.0-294.1) | 288.00 (265.80-313.20) | 290.3 (269-311) |
| Outer Superior | 302.9 (267-336) | 282 (260–308) | 291.7 (263.0-323.0) | 312.89 (289.80-344.40) | 307.3 (286-327) |
| Outer Nasal | 319.0 (280-367) | 298 (272–322) | 302.5 (275.0-331.0) | 316.86 (288.60-354.20) | 323.7 (301-345) |
| Outer Inferior | 294.8 (260-325) | 270 (247–295) | 278.2 (252.5-308.0) | 297.90 (269.60-329.00) | 290.9 (271-310) |

The data are presented as the means ± standard deviations (5th–95th percentiles).

The normative database was provided by Nidek Corporation.

in children[13]. Although we were unable to analyze the effect of age in this study because we included only 8-year-old children, the measurements were still lower than the normative database values for adults included in the OCT database used in this study, suggesting that an increase in retinal thickness occurs with growth[6].

The findings of the present study are consistent with earlier reports in which the average macular thickness was noted to be the greatest in the nasal sector, followed by the superior, inferior, and temporal sectors [11–14]. Similarly, other studies have reported either the nasal or superior sector to be the thickest, while the temporal sector is the thinnest [15,24]. Although there are only a few reports on outer retinal thickness, our finding that the inferior sector was the thinnest agrees with the report of Read and colleagues [11]. They also reported that the NFL was the thickest in the nasal sector and the thinnest in the temporal sector, which was also observed in this study.

Multivariable regression analysis showed that foveal thickness was not significantly correlated with AL but was positively correlated with body height. The foveal thickness increases until approximately puberty due to the elongation of the photoreceptor layer and the increase in the density of cone cells [6,25]. In our study, we observed that the retinal thickness at the central foveal sector increased significantly with increasing body height, which may reflect changes associated with physical growth. Previous studies in children have shown that central and foveal retinal thickness increase or remain unchanged with increased AL and refractive myopia [14,15,22,26]. We believe that AL elongation should be considered in the analysis of retinal thickness in children because it is thought to be confounded by changes in body height [27].

Multivariable regression analysis adjusted for sex and body height showed that the whole, inner, and outer sectors of average macular thickness significantly decreased with increasing AL. In a systematic review of pediatric retinal thickness analyses by Banc *et al*., the association between AL and macular retinal thickness was inconsistent and varied among the reports [26]. Bueno-Gimeno *et al*. reported that the average macular thickness in children was significantly thinner in eyes with a long AL than in eyes with a short AL [28]. Goh *et al*. reported that the thickness of the superior and inferior GC-IPL in children decreased significantly with increasing AL and refractive myopia [29].

Turk and colleagues noted no association between retinal thickness and AL in children [15]. In the present study, there was no correlation with AL at the central sector of average macular thickness, including the fovea, and negative correlations with AL were observed at the inner and outer sectors of average macular thickness. The differences between the present study and these reports may be due to differences in the extent of the macular area analyzed in each report, correction for the ocular magnification effect, racial differences or other factors.

In the present study, the SE showed a weaker correlation with retinal thickness than AL, but some correlations were still observed. In Passani *et al*.'s study, children with myopia were noted to have significantly thinner average macular thickness in seven ETDRS sectors except for the central and nasal inner subfields [14]. Similarly, Al-Haddad reported that the average macular thickness of the outer superior, outer nasal, and outer inferior subfields were significantly thinner in children with myopia [12]. These two reports used cycloplegic drops while the present study did not, which may have caused the difference in the results. Shih *et al*. also reported that at approximately 7–11 years of age, lens thickness decreases in response to axial elongation, thereby maintaining emmetropia [30]. We hypothesized that refractive error, a functional change, may be less involved in the morphological changes in the posterior retina of 8-year-olds because the mechanism of emmetropization in response to axial elongation was affected.

The present study showed that choroidal thickness tended to be negatively correlated with AL and significantly positively correlated with refractive error. In adults, the RPE is shifted posteriorly due to axial elongation, thus reducing the thickness of the choroid, as reported by Jonas *et al* [10]. However, there are limited reports on choroidal thickness in children. Kobia-Acquah and colleagues reported that in children, choroidal thickness was significantly thinner in the parafoveal and perifoveal superior quadrants due to elongation of the AL and myopia [31]. All analyses in this study were performed by one author using the same criteria, and there were 65 participants for whom choroidal thickness could not be measured because the boundaries were not clear, but this was due to unclear OCT images. Although sample bias may have occurred, this inability to measure was not due to choroidal thickness and would have had little effect on the results of the analysis.

Many of the previous reports on children reported that boys had greater retinal thickness than girls [13,22,26]. Zhang *et al*. previously reported that the minimum foveal thickness, inner region thickness, and temporal outer quadrant macular thickness were greater in boys than in girls [32]. The present study revealed similar findings of thicker retinas in boys.

For NFL thickness, we observed that only the outer NFL sectors were thicker in girls than in boys, while Chen and colleagues reported that the NFL was thicker in the inferior and temporal sectors in girls [33]. On the other hand, Huynh *et al*. reported that the NFL in the inferior sector was thicker in boys [22]. It is possible that the significant differences may be a result of the characteristics of the study participants, the number of participants, the instruments used for assessment, participant age, AL, refractive error, and ethnicity.

In cases wherein the right eye could not be used, the participant's left eye was used for measurement. Possible differences between the left and right eyes of the participants were verified, and no significant differences were detected except for in the inner nasal subfield of the NFL (S12–S16 Tables in S1 File).

It was difficult to obtain good cooperation from younger patients during the measurements; hence, the number of rejections was high. Also, sample bias may have affected the results. Compared to the included group, the excluded group had more myopia, longer AL, and poorer uncorrected visual acuity. The narrower range of refractive error and AL in the included group may have affected the results of the analysis. However, sample bias was not considered a significant effect because most of the reasons for exclusion were related to the OCT images, such as maintaining eye gaze, eyelid opening, and posture during the examination. It may also have been influenced by the tight examination schedule, which did not allow enough time for cooperation in eye examinations. It is possible that continued examination in the future during the growth process could provide data.

It has been observed that retinal thickness increases during the first 5 years of life due to morphological development [34]. An 18-month-long longitudinal study on children aged 10–15 years reported that there was little change in the thickness of average macular thickness according to age [11]. To our knowledge, there are no longitudinal studies covering individuals aged 0–18 years. The JECS conducts surveys every 4 years after birth, at ages 4, 8, and 12, with 8-year-olds being included in this study. Thus, the same participants can be surveyed four years later. Since myopia progresses more easily in schoolchildren as the AL elongates [9], we believe that data at the age of 8 years, which is the early stage of schoolchildren in Japan, and the planned survey of changes over time thereafter will be useful in elucidating age-related changes in childhood retinal thickness and the factors that affect the progression of myopia on retinal thickness. As mentioned earlier in this report, the JECS also includes physical measurements, blood tests, physical fitness tests, dental examinations, and questionnaires on parental health status and lifestyle. Future studies on the relationship between these data and the growth of children and their eyes may be able to reveal associations that could not be verified in the present study.

## Conclusions

In this study, we performed retinal thickness measurements in 8-year-old Japanese children. It is possible that changes in retinal structure are influenced by multiple factors, including an increase in AL, physical growth, and sex differences. Although the present study included only 8-year-old children, this study is proposed to continue until 2027 with approximately 4,000 participants, and we will continue to examine longitudinal changes in retinal and choroidal structures in more subjects in the future.

## Supporting information

**S1 File. S1 Table. Comparison of included and excluded groups. S2 Table. A quadrant-by-quadrant comparison for each layer. S3 Table. Average macular thickness. S4 Table. Thickness of the choroid. S5 Table. Thickness of ONL to BM. S6 Table. Thickness of GC-IPL. S7 Table. Thickness of NFL. S8 Figure. Scatter plots of Average macular thickness in all sectors versus AL and body height.** a) AL vs. Fovea, b) AL vs. Average macular thickness whole sector, c) AL vs. Average macular thickness central sector, d) AL vs. Average macular thickness inner sector, e) AL vs. Average macular thickness outer sector, f) Body height vs. Fovea, g) Body height vs. Average macular thickness whole sector, h) Body height vs. Average macular thickness central sector, i) Body height vs. Average macular thickness inner sector, j) Body height vs. Average macular thickness outer sector. **S9 Table: Multivariable regression analysis of the relationship between ONL to BM thickness and axial length, sex, and body height. S10 Table: Multivariable regression analysis of the relationship between GC-IPL thickness and axial length, sex, and body height. S11 Table: Multivariable regression analysis of the relationship between NFL thickness and axial length, sex, and**

body height. S12 Table: Comparison of left and right eye differences in average macular thickness. S13 Table: Comparison of left and right eye differences in choroidal thickness. S14 Table: Comparison of left and right eye differences in ONL to BM thickness. S15 Table: Comparison of left and right eye differences in GC-IPL thickness. S16 Table: Comparison of left and right eye differences in NFL thickness.
(DOCX)

## Acknowledgments

The conclusions of this article are solely the responsibility of the authors and do not represent the official views of the government. We are grateful to all the participants of the JECS and to all individuals involved in the data collection. We also thank the following members of the JECS as of 2024: Zentaro Yamagata, Ryoji Shinohara, Sanae Otawa, Anna Kobayashi, Megumi Kushima, Hideki Yui, Takeshi Inukai, Kyoichiro Tsuchiya, Hirotaka Haro, Masanori Wako, Takahiko Mitsui, Kenji Kashiwagi, Daijyu Sakurai, Koichiro Ueki, Sumire Ono, Tadao Ooka, Hiroshi Yokomichi, Kunio Miyake, Sayaka Horiuchi, and Reiji Kojima.

## Author contributions

**Conceptualization:** Tadao Ooka, Zentaro Yamagata, Kenji Kashiwagi.

**Data curation:** Ryo Harada, Mingxue Bao, Natsuki Okabe, Yuka Kasai, Airi Takahashi, Chio Kuleshov, Yumi Shigemoto, Sanae Otawa, Anna Kobayashi, Megumi Kushima, Kenji Kashiwagi.

**Formal analysis:** Ryo Harada.

**Funding acquisition:** Kenji Kashiwagi.

**Investigation:** Kenji Kashiwagi.

**Methodology:** Ryo Harada, Kenji Kashiwagi.

**Supervision:** Tadao Ooka, Hiroshi Yokomichi, Kunio Miyake, Reiji Kojima, Ryoji Shinohara, Hideki Yui, Sanae Otawa, Zentaro Yamagata, Kenji Kashiwagi.

**Writing – original draft:** Ryo Harada.

**Writing – review & editing:** Mingxue Bao, Natsuki Okabe, Yuka Kasai, Airi Takahashi, Chio Kuleshov, Yumi Shigemoto, Tadao Ooka, Hiroshi Yokomichi, Kunio Miyake, Reiji Kojima, Ryoji Shinohara, Hideki Yui, Sanae Otawa, Anna Kobayashi, Megumi Kushima, Zentaro Yamagata, Kenji Kashiwagi.

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
