## [Decision Letter · Decision Letter 0]

17 Feb 2025

PONE-D-25-01148Retinal Structure and Related Factors in 8-year-old Japanese Children: The Yamanashi Adjunct Study of the Japan Environment and Children's StudyPLOS ONE

Dear Dr. Kashiwagi,

Thank you for submitting your manuscript to PLOS ONE. After careful consideration, we feel that it has merit but does not fully meet PLOS ONE’s publication criteria as it currently stands. Therefore, we invite you to submit a revised version of the manuscript that addresses the points raised during the review process.

We look forward to receiving your revised manuscript.

Kind regards,

Jiro Kogo

Academic Editor

PLOS ONE

**Journal Requirements:**

This adjunct study was supported by Japan Society for the Promotion of Science, No.23K09003.

The conclusions of this article are solely the responsibility of the authors and do not represent the official views of the government. We are grateful to all the participants of the JECS and to all individuals involved in the data collection. We also thank the following members of the JECS as of 2024: 

Zentaro Yamagata, Ryoji Shinohara, Sanae Otawa, Anna Kobayashi, Megumi Kushima, Hideki Yui, Takeshi Inukai, Kyoichiro Tsuchiya, Hirotaka Haro, Masanori Wako, Takahiko Mitsui, Kenji Kashiwagi, Daijyu Sakurai, Koichiro Ueki, Sumire Ono, Tadao Ooka, Hiroshi Yokomichi, Kunio Miyake, Sayaka Horiuchi, and Reiji Kojima.

The Japan Environment and Children’s Study was funded by the Ministry of the Environment, Japan. This adjunct study was supported by Japan Society for the Promotion of Science, No.23K09003.

This adjunct study was supported by Japan Society for the Promotion of Science, No.23K09003.

"NONE"

5. We note that your Data Availability Statement is currently as follows: All relevant data are within the manuscript and its Supporting Information files.

Reviewers' comments:

Reviewer's Responses to Questions

**Comments to the Author**

1. Is the manuscript technically sound, and do the data support the conclusions?

Reviewer #1: Partly

Reviewer #2: Yes

2. Has the statistical analysis been performed appropriately and rigorously? 

Reviewer #1: Yes

Reviewer #2: Yes

3. Have the authors made all data underlying the findings in their manuscript fully available?

Reviewer #1: Yes

Reviewer #2: Yes

4. Is the manuscript presented in an intelligible fashion and written in standard English?

Reviewer #1: Yes

Reviewer #2: Yes

5. Review Comments to the Author

**Reviewer #1: ** The authors reported retinal structure and related factors in 8-year-old Japanese children. Although this manuscript has some interesting observations, the manuscript needs to be improved before it can be accepted in the journal.

Introduction

Line 60: Space is needed between sentences

Figure 1

Why do the authors exclude the nerve fiber layer and inner nuclear layer from the inner retinal layer, and the outer plexiform layer from the outer retinal layer? Isn't the inner retinal layer that the authors refer to GCIPL (ganglion cell and inner plexiform layer)? The terms used in the paper and the terms used by the authors have different meanings. The term All retinal layers (ARLs) is also not used in the paper, so please change it to the term used in ordinary papers.

At 8.5 years of age, the choroid is very thick and many eyes cannot be measured, so how did the authors measure the choroid? The boundary between the sclera and choroid is not clear in Figure 1, so please provide an image that clearly shows the boundary. If there are many cases where the choroid cannot be measured, please explain the impact on the results.

Inclusion criteria

Please state the number of subjects who used their left eye.

Table 2

What does "Choroid only" mean? Does it mean that the choroid was measurable in 239 eyes?

Discussion

Line 278: This study is part of a cohort survey, but as an ophthalmology study it is a cross-sectional study, so please correct this.

**Reviewer #2:**  Main comments

1. justification for selecting 8-year-olds

Further clarification is needed as to why the 8-year-olds were chosen as the study subjects. Providing a clear justification for why the 8-year-olds were chosen would strengthen the methodological foundation of the study. 2.

2. possible underestimation due to exclusion bias

The manuscript states that a significant number of participants were excluded due to poor OCT image quality, lack of cooperation in examination, and suboptimal fixation. Notably, the excluded group had significantly longer axial lengths and a higher prevalence of myopia. This exclusion may have underestimated the association between axial length and retinal thickness in this study. For the sake of transparency, it would be beneficial to discuss in more detail the potential impact of this exclusion bias on the study results.

Minor comments.

1. the circled numbers in Figure 1b are not shown.

2. it is difficult to see the relationship between the columns of independent variables and dependent variables in Table 6.

3. a brief description of the specifications of the NIDEK RS-3000 Advance used for the measurements (e.g., spatial resolution, measurement accuracy, etc.) would facilitate comparison with other studies.

4. a colon is missing at the end of the sentence in line82.

6. PLOS authors have the option to publish the peer review history of their article (what does this mean? ). If published, this will include your full peer review and any attached files.

**Do you want your identity to be public for this peer review?** For information about this choice, including consent withdrawal, please see our Privacy Policy .

Reviewer #1: No

Reviewer #2: No

---

## [Author Response · Author response to Decision Letter 1]

8 Apr 2025

Thank you for reviewing our manuscript.

Our comments to the reviewers are as follows

Our response is in italics after the reviewer's comments.

Reviewer #1: The authors reported retinal structure and related factors in 8-year-old Japanese children. Although this manuscript has some interesting observations, the manuscript needs to be improved before it can be accepted in the journal.

Introduction

Line 60: Space is needed between sentences

I have corrected that.

Figure 1

Why do the authors exclude the nerve fiber layer and inner nuclear layer from the inner retinal layer, and the outer plexiform layer from the outer retinal layer? Isn't the inner retinal layer that the authors refer to GCIPL (ganglion cell and inner plexiform layer)? The terms used in the paper and the terms used by the authors have different meanings. The term All retinal layers (ARLs) is also not used in the paper, so please change it to the term used in ordinary papers.

The retinal layers were changed as follows.

ARL→average macular thickness

IRL→GC-IPL

ORL→ONL to BM

Note that the INL and OPL were removed from the analysis because the retinal ganglion cells in the inner retina and the photoreceptor layer in the outer retina were the primary targets of investigation because they are commonly used for disease monitoring and therefore have high clinical significance.

At 8.5 years of age, the choroid is very thick and many eyes cannot be measured, so how did the authors measure the choroid? The boundary between the sclera and choroid is not clear in Figure 1, so please provide an image that clearly shows the boundary. If there are many cases where the choroid cannot be measured, please explain the impact on the results.

Figure 1 was changed to a clearer image.

The choroid was measured using the border between the sclera and the choroid as the boundary between the lower part of the OCT image, where there is no image depiction, and the area where there is an image depiction, albeit indistinct.

All analyses were performed by one author using the same criteria. There were 65 participants for whom choroidal thickness could not be measured because the boundaries were not clear, but this was because the OCT images themselves were unclear. It is possible that sample bias may have occurred, but the exclusion was not due to thick choroidal thickness, which would have had little effect on the results of the analysis.

This has been added to the text.

Inclusion criteria

Please state the number of subjects who used their left eye.

I added that 186 were in the right eye and 118 in the left eye.

Table 2

What does "Choroid only" mean? Does it mean that the choroid was measurable in 239 eyes?

I added that only the choroid was measurable in 239 eyes.

Discussion

Line 278: This study is part of a cohort survey, but as an ophthalmology study it is a cross-sectional study, so please correct this.

Corrected to cross-sectional study.

Reviewer #2: Main comments

1. justification for selecting 8-year-olds

Further clarification is needed as to why the 8-year-olds were chosen as the study subjects. Providing a clear justification for why the 8-year-olds were chosen would strengthen the methodological foundation of the study. 2.

The JECS conducts surveys every 4 years after birth, at ages 4, 8, and 12, with 8-year-olds being included in this study. Thus, the same participants can be surveyed four years later. Since myopia progresses more easily in schoolchildren as the AL elongates, we believe that data at the age of 8 years, which is the early stage of schoolchildren in Japan, and the planned survey of changes over time thereafter will be useful in elucidating age-related changes in childhood retinal thickness and the factors that affect the progression of myopia on retinal thickness.

This has been added to the text.

2. possible underestimation due to exclusion bias

The manuscript states that a significant number of participants were excluded due to poor OCT image quality, lack of cooperation in examination, and suboptimal fixation. Notably, the excluded group had significantly longer axial lengths and a higher prevalence of myopia. This exclusion may have underestimated the association between axial length and retinal thickness in this study. For the sake of transparency, it would be beneficial to discuss in more detail the potential impact of this exclusion bias on the study results.

Compared to the included group, the excluded group had more myopia, longer AL, and poorer uncorrected visual acuity. The narrower range of refractive error and AL in the included group may have affected the results of the analysis. However, since most of the reasons for exclusion were related to OCT imaging, such as maintaining eye gaze, open eyelids, and posture during the examination, the exclusion bias was not considered to be a significant effect.

Added to the text.

Minor comments.

1.the circled numbers in Figure 1b are not shown.

Fig 1b has been corrected.

2. it is difficult to see the relationship between the columns of independent variables and dependent variables in Table 6.

The structure of Table 6 has been changed.

3. a brief description of the specifications of the NIDEK RS-3000 Advance used for the measurements (e.g., spatial resolution, measurement accuracy, etc.) would facilitate comparison with other studies.

Added to the text.

4. a colon is missing at the end of the sentence in line82.

Added to the text.

---

## [Editor Report · Decision Letter 1]

14 Apr 2025

Retinal Structure and Related Factors in 8-year-old Japanese Children: The Yamanashi Adjunct Study of the Japan Environment and Children's Study

PONE-D-25-01148R1

Dear Prof. Kahiwagi

We’re pleased to inform you that your manuscript has been judged scientifically suitable for publication and will be formally accepted for publication once it meets all outstanding technical requirements.

Kind regards,

Jiro Kogo

Academic Editor

PLOS ONE

---

## [Editor Report · Acceptance letter]

PONE-D-25-01148R1

PLOS ONE

Dear Dr. Kashiwagi,

I'm pleased to inform you that your manuscript has been deemed suitable for publication in PLOS ONE. Congratulations! Your manuscript is now being handed over to our production team.

Kind regards,

on behalf of

Prof. Jiro Kogo

Academic Editor

PLOS ONE